# Comparative Structure Analysis of the Multi-Domain, Cell Envelope Proteases of Lactic Acid Bacteria

**DOI:** 10.3390/microorganisms11092256

**Published:** 2023-09-08

**Authors:** Lise Friis Christensen, Magnus Haraldson Høie, Claus Heiner Bang-Berthelsen, Paolo Marcatili, Egon Bech Hansen

**Affiliations:** 1National Food Institute, Technical University of Denmark, Kemitorvet, DK-2800 Kongens Lyngby, Denmark; 2Department of Health Technology, Technical University of Denmark, Ørsteds Plads, DK-2800 Kongens Lyngby, Denmark

**Keywords:** cell envelope protease, subtilase, lactic acid bacteria, protein structure, AlphaFold 2 models, extracellular proteolysis

## Abstract

Lactic acid bacteria (LAB) have an extracellular proteolytic system that includes a multi-domain, cell envelope protease (CEP) with a subtilisin homologous protease domain. These CEPs have different proteolytic activities despite having similar protein sequences. Structural characterization has previously been limited to CEP homologs of dairy- and human-derived LAB strains, excluding CEPs of plant-derived LAB strains. CEP structures are a challenge to determine experimentally due to their large size and attachment to the cell envelope. This study aims to clarify the prevalence and structural diversity of CEPs by using the structure prediction software AlphaFold 2. Domain boundaries are clarified based on a comparative analysis of 21 three-dimensional structures, revealing novel domain architectures of CEP homologs that are not necessarily restricted to specific LAB species or ecological niches. The C-terminal flanking region of the protease domain is divided into fibronectin type-III-like domains with various structural traits. The analysis also emphasizes the existence of two distinct domains for cell envelope attachment that are preceded by an intrinsically disordered cell wall spanning domain. The domain variants and their combinations provide CEPs with different stability, proteolytic activity, and potentially adhesive properties, making CEPs targets for steering proteolytic activity with relevance for both food development and human health.

## 1. Introduction

Lactic acid bacteria (LAB) are a heterogeneous group of Gram-positive bacteria that play important roles in both food development and human health [1,2]. LAB are found in a wide range of nutrient-rich ecological niches and can be divided into the six families *Aerococcaceae, Carnobacteriaceae, Enterococcaceae, Lactobacillaceae, Leuconostocaceae,* and *Streptococcaceae* [1,2].

During adaptation, extensive genome reductions have occurred resulting in the fastidious growth requirements of LAB [3]. Dairy-derived LAB strains are characterized by a high number of amino acid (aa) auxotrophies and the presence of a proteolytic system for milk protein utilization to meet their aa requirements [2,4]. The utilization of environmental proteins, such as the milk protein casein, is initiated by an extracellular cell envelope protease (CEP) that degrades proteins into peptides. Specific peptide transporters allow the import of aa and peptides into the bacterial cell, in which the hydrolysis of the peptides into aa is completed by an interplay of various intracellular peptidases. CEPs are typical large serine proteases that belong to the protease subfamily S8A of subtilisin-like serine proteases or subtilases [2,5].

These extracellular proteases contribute to the proteolytic phenotypes of LAB strains, though they are not found in all LAB strains [6,7,8]. The dairy industry has utilized LAB strains with different proteolytic phenotypes to steer the speed of acidification and flavor formation [2,4]. CEPs from dairy-derived LAB strains display different stability and proteolytic activity against casein. The different proteolytic activity of CEP homologs from *Lactococcus lactis* strains has been particularly studied. These proteases, known as PrtP, have different substrate selectivity and specificity, despite having high protein sequence identities [9,10]. Genome analysis of lactobacilli has identified several PrtP homologous genes, encoding proteins with sequence identities between 20% and 100% [6]. PrtP from *Lacticaseibacillus paracasei* subsp. *paracasei* NCDO151 has been characterized [11], as well as PrtB and PrtL from *Lactobacillus delbrueckii* subsp. *bulgaricus* and *Lactobacillus delbrueckii* subsp. *lactis*, respectively [12,13]. *Lactobacillus helveticus* has different homologous CEPs encoding genes, of which *L. helveticus* CNRZ32 has the four CEP homologs PrtH, PrtH2, PrtH3, and PrtH4. PrtR is another characterized *Lactobacillus* CEP homolog, which is caseinolytic, though it is from a human vaginal isolate *Lacticaseibacillus rhamnosus* BGT10 [14]. CEPs from dairy-derived *Streptococcus thermophilus* strains have also been studied, and they are known as PrtS [7,15]. Pathogenic streptococci also encode CEP homologs, such as the two *Streptococcus pyogenes* CEPs (SpyCEPs) ScpA and ScpC [16,17,18]. ScpA and ScpC display exceptional substrate specificity and selectivity against anaphylatoxins and chemokines in the human immune response. These CEPs, along with other CEPs that are homologous with PrtP, will from now on be referred to as PrtP homologs. If necessary, LAB stains are added to the name of the PrtP homologs to distinguish between PrtP homologs from different strains such PrtP_Wg2_ and PrtP_SK11_ from *L. lactis* Wg2 and SK11, respectively.

Comparative sequence studies of PrtP homologs from diary- and human-derived LAB strains have shown that PrtP homologs share multiple functional domains [2,4,5]. The pre-pro (PP) domain precedes the N-terminus of the protease (PR) domain, and it is responsible for the proper secretion and activation of the protease. The PR-domain is defined as the region with sequence homology to the catalytic domain of the *Bacillus* protease subtilisin, which is characterized as the subtilase family S8 protease domain in the protein families’ database Pfam [5,19]. Some PrtP homologs have a protease-associated (PA)-domain inserted in the PR-domain [5]. A large A-domain region flanks the C-terminus of the PR-domain in all PrtP homologs. In most PrtP homologs, the A-domain region is followed by a B-domain region of different lengths [2,4,5]. Sometimes the B-domain is followed by a helix (H) spacer domain [5]. All PrtP homologs have been reported to contain a cell wall (W) spacer domain [2,4]. The W-domain of some PrtP homologs comes before a typical cell wall anchor (AN), whereas some *Lactobacillus* PrtP homologs have a surface layer attachment domain included in their W-domains [2]. Domain boundaries are determined by sequence homologies [5], though low sequence identities make domain delimitation more elusive, as seen in the boundaries between the A- and B-domain regions, as well as in the W-domain.

Protein structures can guide the fundamental understanding of protein function. The experimental determination of the entire protein structure of CEPs is challenging and time-consuming due to the size of CEPs and their cell-envelope attachment. However, protein crystal structures of SpyCEPs have been resolved, showing that the A- and B-domain regions contain repeating β-sandwich domain structures [16,17,20]. The A-domain region contains three fibronectin type-III-like domain structures, namely Fn1, Fn2, and Fn3, of which Fn2 plays a role in substrate interaction. Homology modeling has provided structural insight into the *Lactococcus lactis* PrtP homologs, using subtilisin and ScpA structures as templates [5,10]. A combined approach using different homology modeling tools achieved the first three-dimensional model of an entire *L. lactis* PrtP, which is from *L. lactis* MS22337 derived from spontaneously acidified camel milk [21]. This structural model implicates the bacterial interaction with casein micelles. The structures of a few PrtP homologs have recently been predicted by the neural network-based algorithm AlphaFold (AF) that demonstrates accuracy comparable with experimentally determined protein structures, greatly outperforming other structure prediction methods [22]. The AF-predicted structures are available in the continuously growing AlphaFold Protein Structure Database that covers parts of the catalogued proteins in the Universal Protein Resource (UniProt) archive [23]. The high accuracy and recent accessibility of AF protein structure models inspire and facilitate a comparative structure analysis of diverse PrtP homologs.

The growing interest in the application potential of proteolytic LAB strains, including their extracellular hydrolysis of plant proteins, has been facilitated by the demand for plant-based food products [24]. However, comparative structure analysis of PrtP homologs has been limited to homologs from dairy and human LAB isolates [2,4,5,12], despite the presence of PrtP homologous genes in plant-derived LAB strains [8]. The current study aims to clarify the prevalence and structural diversity of PrtP homologs among LAB species from different natural habitats. PrtP homologs from plant-derived LAB species are identified by a homology search. These homologs are compared to previously characterized PrtP homologs from LAB strains derived from both dairy and human isolates. In addition to SpyCEP crystal structures, AF modeling provides a selection of three-dimensional PrtP homologous structures. The comparative structure analysis specifies important domain boundaries and novel domain architectures of PrtP homologs of different LAB species from dairy, human, and plant ecological niches.

## 2. Materials and Methods

### 2.1. Protease Homology Search

Protein sequences of PrtP homologs were collected from UniProtKB and from GenBank at NCBI (Table 1). PrtP homologs from plant-originated LAB strains were acquired by a tBLAST search using the protein query sequences of PrtP_Wg2_ (UniProtKB ID: P16271) and subtilisin (UniPrtKB ID: P00780). The CLC Main Workbench by QIAGEN [25] served as platform for a tBLAST search against the continuously expanding National Food Institute Culture Collection (NFICC) at the Technical University of Denmark. The NFICC contained whole genome sequence data of 331 LAB strains that were isolated from a broad range of natural plant sources and pre-fermented plant-based food matrices. Species identification was based on MALDI-TOF mass spectrometry, and sequence data of the whole genomes were stored as unassembled contigs. Sequence proteins from LAB strains with known plant origin were included for further analysis if the E-values of the tBLAST search were less than 0.01. Extracellular signal peptides were predicted using the SignalP 6.0 server with “Organism” set to “Other” and “Model mode” set to “slow” (Table 2) [26]. Protein sequences with signal peptide scores above 0.01 were analyzed for PrtP_Wg2_ homology, using HHpred pairwise comparison with default settings as part of the MPI Bioinformatics Toolkit (version: 57c8707149031cc9f8edceba362c71a3762bdbf8) [27]. HHpred was also used to search for homologs in the structure database PDB_mmCIF70_31_Jul.

### 2.2. Properties of Protein Sequence Homologs

The MAFFT algorithm as part of the MPI Bioinformatics Toolkit was used to generate MSAs of the protein sequences [29]. The CLC Main Workbench was used for sequence analyses, including construction of phylogenetic trees using the MSAs and Neighbor Joining method. Protein distances were calculated using Jukes–Cantor, and bootstrapping was used to assess the reliability of the inferred trees, with default bootstrap value set to 100.

Protein sequences were classified into known domain regions by searching against the Pfam database [19]. Potential tandem repeat motifs of selected protein regions were identified using the web interfaces T-REKS and XSTREAM with default settings (Table 2) [31,32]. ProtParam was used to compute aa composition and theoretical pI of protein sequences [33].

### 2.3. Protein Structure Modeling

AF modeling using AlphaFold 2 was used to create three-dimensional structure models where the input was the PrtP protein sequences without predicted signal peptides (Table 2) [22,26]. Predicted signal peptides were trimmed using SignalP 6.0, with the parameter “Organism” set to “Other” and “Model mode” set to “slow”. We predicted the structure models using the ColabFold implementation of AlphaFold 2, with default parameters [28].

PyMOL 2.5 by Scrödinger was used for protein structure analysis and visualization. The NetSurfP-3.0 tool was used for supplementary prediction of protein structural features including secondary structures and protein ID [30].

## 3. Results

### 3.1. Phylogenetic Clustering of Diverse PrtP Homologs

A search against the NFICC genome database for PrtP homologs identified 27 protein sequences which have a sequence homology to PrtP_Wg2_. These homologies were supported by HHpred pairwise comparisons, predicting 100% homology probabilities based on E-values below 10^−37^ (Table 3, Appendix A). The PrtP homologs had 1454–1913 amino acids (aa) and provided sequence coverages above 50% of PrtP_Wg2_. These PrtP homologs were from 25 LAB strains that were isolated from sourdough or natural plant habitats such as berry, evergreen, flower, fruit, seashore plant, seaweed, and tap/tuber roots (Appendix A). The LAB strains represent four genera including the six LAB species *Carnobacterium maltaromaticum*, *Enterococcus durans*, *Leuconostoc lactis*, *Leuconostoc mesenteroides*, *Leuconostoc pseudomesenteroides*, and *Pediococcus pentosaceus*. Most of the identified PrtP homologs were from one of the three *Leuconostoc* species where three PrtP homologs were identified in the same *Leuconostoc pseudomesenteroides* strain NFICC96. These three protein homologs displayed broad sequence similarities among the identified PrtP homologs from 30–96% identity with PrtP_Wg2_ (Appendix A).

These plant-associated PrtP homologs were analyzed together with the PrtP homologs in Table 1, including PrtP homologs from dairy- and human-derived LAB strains. The 45 PrtP homologous protein sequences were aligned, showing high sequence conservation within their PR-domains. The flanking regions of the PR-domain showed larger sequence variations among the PrtP homologs where sequence coverage decreased along the C-terminal part of the multiple sequence alignment (MSA). The protease domain region, including PR- and PA-domains, was used for the phylogenetic analysis of the PrtP homologs, which divided the PrtP homologs into 12 overall clusters (I-XII) besides the cluster constituted by subtilisin (XIII) (Figure 1). This phylogenetic tree shared the same overall tree typology as the tree which was generated using the whole protein sequences (Appendix A). The phylogenetic clustering of the PrtP homologs was not necessarily determined by the origins of the LAB strains. PrtP homologs with the same LAB species or genera often clustered in the same cluster, though larger evolutionary distances sometimes appeared between PrtP homologs of the same LAB strain. This was observed for the three and four PrtP homologs of *Leuconostoc pseudomesenteroides* NFICC96 and *Lactobacillus helveticus* CNRZ32, respectively.

### 3.2. Multi-Domain Structures of PrtP Homologous Proteins

Among the 45 PrtP homologs, the crystal protein structures were available for ScpA, ScpC, and subtilisin. Subtilisin’s crystal structure corresponded to the recently available AF model (AF-P00780) [23], but the AF model included the propeptide region. In a comparative structure analysis of PrtP homologs, the AF model of subtilisin was included as a reference protein structure, along with the PDB structures of ScpA and ScpC (Table 1). The analysis included additional AF models of 18 PrtP homologs to represent the 12 PrtP clusters, LAB species, and LAB strains from various habitats (Figure 1). The PrtP homologs were selected to represent evolutionary close and distance protein homologs where Cluster V, VI, X, and XII were represented by more than one PrtP homolog. Cluster X was represented by five PrtP homologs, including some of the most studied dairy-associated homologs PrtP_SK11_, PrtP_Wg2_, and PrtP_MS22337_, which may shed light on the function of small structural variations.

Each of the 18 AF models were selected among the five ranked AF structure models. Usually, the best AF-ranked structure was selected, however, lower ranked AF models were used if the propeptide domain showed position errors lower than expected (Appendix A). The MSA used for the AF structure modeling showed low evolutionary information of the query PrtP homologous sequences (Appendix A). Here, the sequence coverages of the AF models were generally low and with sequence identities below 0.5. The sequence coverages were best around the catalytic domain regions of the PrtP homologs and reduced along their flanking N-termini and C-termini. The predicted local distance difference test (pLDDT) scores, which indicated the local residue structural quality, fluctuated along the sequences, with values generally above 70 (Appendix A). This indicated that most regions of the predicted structures had good protein backbone predictions [22]. Some regions were predicted with an even higher structural accuracy, supporting the positions of aa side-chains (pLDDT > 90). Other regions, which were generally short, were predicted with low confidence as pLDDT fell below 70. Low confidence regions appeared mostly in the terminal ends of the PrtP homologous sequences, reflecting the limited evolutionary information in these regions of the MSAs (Appendix A). This included the furthest N-termini of the protease sequences and regions within their C-termini. Additionally, longer regions with pLDDT below 50 appeared along the C-termini and in the furthest N- and C-termini of some protease sequences. These low-confidence regions were not interpreted based on the AF models alone, but compared with the secondary structure and protein intrinsic disorder (ID) predictions of NetSurfP-3.0. The qualities of the AF models were too low to approach the domain–domain interactions within the large proteins (Appendix A). However, the AF models supported protein domain delimitation, with the protein domains of the AF models appearing as distinct structural units.

The PR-domain was defined as the S8 protease domain of subtilisin [5], whereas the PA-domain appeared as an insert domain dividing the PR-domain into the PR1 and PR2 regions. The LPXTG cell wall anchor (AN) domain contained the canonical LPXTG motif or a variant of it. The domain delimitation was compared to identified Pfam domains, including the Subtilase protease (PR) domain (PF00082), the protease-associated (PA)-domain (PF02225), the fibronectin type-III (Fn) domain (PF06280), the LPXTG cell wall anchor (AN) domain (PF00746), and the surface layer protein A (SlpA) anchoring domain (PF03217). Several domains were not identified as Pfam domains. The boundary between SS and the propeptide of the PP domain was generally identified by SignalP. The propeptide was determined as the region between the predicted prepeptide cleavage site and the beginning of the PR-domain. The neighboring C-terminal domain to PR-domain was the only Fn-domain identified by Pfam. In ScpA and ScpC, three domains in the A-domain region had previously been identified as three fibronectin type-III-like domains and were designated as Fn1, Fn2, and Fn3 [16,17]. We continued this Fn nomenclature of the domains in the B-domain region. Additionally, the H-domains and the W-domains were determined based on structural characteristics.

The protein domain delimitation revealed 11 overall domain architectures of the PrtP homologous proteins, with subtilisin being the simplest, containing a PR-domain in addition to the N-terminal pre-pro (PP) peptide domain region (Figure 2). Within each of the phylogenetic clusters (Figure 1) the domain architecture was shared by all members of the group. Cluster V, VII, and VIII shared the same domain architecture, whereas all other clusters showed different architectures, giving a total of 10 different domain compositions for the PrtP homologs of the LAB. The PrtP homologs from LAB had different lengths between 1454 and 2020 aa, while subtilisin had only 379 aa. Subtilisin differed from the other PrtP homologs by the absence of the A-domain region constituted by the Fn1, Fn2, and Fn3 domains. The domain architecture variations were mostly apparent within the C-terminal part of the PrtP homologs. ScpA did not have a B-domain region, whereas the B-domain region had two-to-seven Fn-domains (Fn4–Fn10). Some PrtP homologs contained a helical region. Two different cell wall attachment domains were observed among the PrtP homologs, which had either an AN-domain or a SlpA-domain. The PrtP_NFICC96H_ was characterized by the absence of both a W-domain and an attachment domain. Hereby, the W-domain was only observed in relation to the cell wall attachment domains. Some PrtP homologs differed from each other by the presence or absence of an inserted PA-domain. However, all PrtP homologs had a PP-domain region constituted by a prepeptide and a propeptide.

### 3.3. Structural Characteristics of the Distinctive Domains

#### 3.3.1. Pre-Propeptide Domain

As most subtilisin-like proteases, PrtP homologs are synthesized as pre-proenzymes and first activated by autocatalytic cleavage of the PP-domain region after translocation across the cell membrane.

The majority of PrtP homologs each had a prepeptide that encoded a typical predicted secretory signal (SS) peptide for translocation over the cell envelope. SignalP did not identify a clear SS peptide for PrtP_NFICC96H_ (Appendix A), though its prepeptide region shared sequence characteristics with the other predicted tripartite SS peptides of the PrtP homologs, containing a hydrophobic core region and two polar flanking regions. The length of the prepeptides varied from 24 to 64 residues, of which the N-terminal polar flanking region mostly contributed to the variation in the lengths, similar to other SS peptides [34]. A cleavage site for the signal peptidase I was predicted at the end of the prepeptide regions. Signal peptidase I releases translocated pre-proteins from the cytoplasmic site to the cell envelope, whereas non-cleavable N-terminal SS peptides can serve as cytoplasmic membrane anchors [35]. Cleavage after the prepeptide of PrtP homologs has, to our knowledge, not been elucidated as the prepeptide region of the PrtP homologs which might be removed as part of the autonomous removal, together with the propeptide in the PP-domain.

The propeptides could be divided into two structural groups as the propeptides were predicted to be either ID regions or structured regions. These structural differences of the propeptides were also reflected by different pLDDT of the AF models. Here, regions with low pLDDT were indicted to be ID regions given that long regions with pLDDT < 50 can be interpreted as a prediction of ID [36]. The ID characterization of the propeptides were corroborated by NetSurfP-3.0 predictions. The ID propeptides included ScpA, PrtS, ScpC, PrtH2, PrtR, PrtP_NFICC80_, and PrtP_NFICC200_, which represented Clusters I-III, XI, and XII (Figure 3). These propeptides differed in length with 48–147 aa and with no apparent conserved sequence similarities. On average, the ID propeptides had 20% charged residues (range 10–32%), of which 15% were acidic (Asp, Glu; 4–27%) and 5% were basic (Arg, Lys, His; 0.9–9%). The content of Pro residues was relatively high with 7% in the range 4–14% while Cys was absent. The structured propeptides were represented by the other 14 PrtP homologs, including subtilisin. Compared to the ID propeptides, the structured propeptides had higher pLDDT and were generally longer, with 145–172 aa. The average content of charged residues was 25% (17–39%), of which the acidic and basic residue contents were, on average, 11% (8–21%) and 13% (9–22%), respectively. The content of Pro residues was 3% on average (1–6%), whereas Cys residues were absent. The structured propeptides had a conserved structural fold (RMSD < 2.1 Å) that was composed of β1-α1-β2-β3-α2-β4 (Figure 3). The four β-strands formed an antiparallel β-sheet, while the two helices were antiparallel. α2 was the shortest helix, with two turns, whereas α1 had generally seven-to-eight turns, except for α1 in the propeptide of subtilisin. The propeptide of subtilisin shared a 14–25% sequence identity with the other homologous propeptides that had a 24–58% sequence identity across Cluster IV, V, VI, IX, and X. Except for the short propeptides in subtilisin, the conserved core of the structured propeptides had a N-terminal helix region with 50–65 aa. This region was less conserved, with a 6–38% sequence identity between clusters, adopting different helical structures.

The structured propeptides formed complexes with the PR-domains in the AF models of PrtP_NFICC96H_, PrtH, PrtB, and PrtL. The conserved fold established an interaction surface to the PR-domain by the N-termini of α1 and by the beta-sheet. The α1 helix interacted with a three-turn α helix of PR. This α helix was absent in subtilisin, which also had a shorter α1 in its propeptide. The β-sheet packed against the N-termini of two parallel helices in the PR-domain. Hereby, β4 directed an unstructured protein region into the catalytic cleft of PR where the catalytic tirade was in spatial proximity to the cleavage site (Figure 3). The cleavage sites of PrtP homologs had been proposed to follow the motif pattern KVY[YH][PA][TN]↓D [5], but this motif was not conserved across clusters of the PrtP homologs. Other aa constraints might apply to this region, which appeared to have a distinct length of seven aa between β4 to the cleavage site (Figure 3).

The conserved structure of the propeptides appeared to dictate the cleavage site for the autocatalytic processing step, removing the propeptide from the mature PrtP homologs. In contrast, the lack of a fixed tertiary fold of the ID propeptides and the lack of the conserved cleavage motif suggested a less distinct cleavage site for propeptides characterized by ID as observed for ScpA. The propeptide of ScpA contained several processing sites for hydrolysis by either autocatalytic intramolecular cleavage or by exogenous protease activity [37]. The propeptide of ScpC had been experimentally verified to be characterized by ID, but this region was also displaying significant helical propensity, which could be important for the secretion and folding processes of ScpC [20]. However, the structural differences of the propeptides among PrtP homologs indicated that PrtP homologs might follow different maturation processes guided by the structural characteristics of their propeptides.

#### 3.3.2. Catalytic Region

The catalytic region followed the N-terminal PP-domain and consisted of a PR-domain with or without an insert PA-domain (Figure 2). The conserved PR-domain of the PrtP homologs had a well-defined alpha/beta fold containing a 7-stranded parallel β-sheet that characterized the subtilase family. PR was the most conserved domain, with a 25–69% sequence identity across Clusters I–XIII. The PA-domain divided the PR-domain into two regions, PR1 and PR2, which were equally conserved. PR1 had 231–301 aa and was larger than the PR2 region with 114–132 aa (Figure 2). Together the PR1 and PR2 regions formed a domain with 347–416 aa, similar to the PR-domains with no insert domains of 362–370 aa. The model structures aligned well when superimposed on the PR-domain of PrtP_MS22337_ (RMSD < 1.2 Å). The PR-domain of subtilisin differed from the other PR-domain by its smaller size of 274 aa and the absence of a tree-turned helix structure PR-domain (Figure 4). This short helix structure was surface exposed and might interact with the propeptide during the folding process as discussed above (Figure 3). The catalytic triads, with Asp and His in the PR1 region and Ser in the PR2 region, were spatially close to each other, with distances of approximately 7.5, 8.4, and 10 Å between the C_α_-atoms in Asp/His, His/Ser, and Asp/Ser, respectively.

The catalytic region of the PrtP homologs from Clusters XI-XIII, including subtilisin, lacked a PA-domain. The remaining ten clusters contained PrtP homologs with a PA-domain ranging from 137 to 189 aa. The PA-domains had a 9–55% sequence identity across clusters, by which PA had an up to 45% lower sequence identity than the rest of the catalytic region. The PA-domain showed its structure homology when superimposed on the PA-domain of PrtP_MS22337_. Most RMSD-values were below 2.5 Å, except for the PA-domains of ScpC and PrtP_NFICC96H_, which had RMSD-values of 3.0 Å and 3.3 Å, respectively. Secondary structures were not well defined in the PA-domain of ScpC. The other PA-domains shared a common core fold β1-β2-β3-α1-β4-β5-α2-β6 in which β1 and β6 formed an antiparallel β-sheet, marking the N- and C-terminal boundaries of the PA-domain. This antiparallel β-sheet was distinctive for all PA-domains, though the lengths of the β-strands varied. A parallel β-sheet made the core, including β2, β3, β4, and β5, which was surrounded by two peripheral helices α1 and α2 (Figure 4). Other adjacent but less defined secondary structures were present in some PA-domains, such as PrtP_NFICC96H_. The PA-domain of PrtP_NFICC96H_ had 41–52 more aa residues compared to its homologs, extending the loop between α2 and β6 with some helical structure.

#### 3.3.3. Tail of Fibronectin-like Domains

Following the PR-domain, the A- and B-domain regions formed a C-terminal tail of domains with predominating β-strand structures (Figure 2). Subtilisin and ScpA were different from the other PrtP homologs. Subtilisin lacked both the A- and B-domain regions, and ScpA lacked the B-domain region. The A-domain regions had always three fibronectin type-III-like domains (Fn1–Fn3), whereas the B-domain regions had two-to-seven (Fn4–Fn10) (Figure 2). For all PrtP homologs, Fn1 was the only Fn-domain, which was identified as the Pfam domain Fn3_5. This fibronectin type-III-like domain lacked disulfide bonds and had a structural core with a sandwich fold. This fold was made up of two antiparallel β-sheets, one with three strands and one with four, and with the N- and C-termini oriented at opposite ends [16]. Fn2 was generally the largest of the Fn-domains, containing 153–217 aa. The other Fn-domains had 73–164 aa.

The Fn-domains were compared intra- and intermolecularly by generating a phylogenetic tree based on an MSA of all Fn-protein domain sequences (Figure 5). All Fn1 and Fn2 domains were in two distinct phylogenetic clusters, indicating that these domains had conserved functionality among PrtP homologs. The other Fn-domains had a tendency to cluster according to their numbers, but some Fn-domains were outliers and did not follow this pattern in their clustering. It should be noted that not all branches had a bootstrap support above 70% (Figure 5). However, the general clustering pattern suggested that the Fn-domains were conserved across and not within the PrtP homologs and that Fn-domains along the fibronectin-like tails of the PrtP homologs were different from each other. An intramolecular comparison of the Fn model structures in PrtP_MS22337_ (RMSD 1.9–18 Å) and ScpC (RMSD 1.6–18 Å) also indicated structural differences between the Fn-domains, but some Fn-domains were also suggested to share structure homologies (Appendix A). In order to clarify the structural characteristics of the A- and B-domain regions, the structures of the Fn-domains were superimposed according to their designated numbers (Figure 2) and to their clustering patterns in the phylogenetic tree (Figure 5).

The Fn1 domains had 124–168 aa and showed their structure homology when superimposed on PrtP_MS22337_ (RMSD < 3.0 Å) (Figure 6A). However, the sequence identity was down to 15%. The sandwich fold of the Fn1 domain corresponded to the fibronectin type-III-like domain. The homologous Fn1 domains had structural variations in the regions Fn1_β3–β4_ and Fn1_β5–β6_. Fn1_β3–β4_ had a β-hairpin structure in the PrtP homologs, except for ScpA, PrtS, ScpC, and PrtP_NFICC96H_. PrtP_NFICC96H_ had an extended loop whereas ScpA, PrtS, and ScpC possessed shorter linker regions. Fn1_β3–β4_ was oriented towards Fn2, to which Fn1 formed an interface. In ScpA, PrtS, and ScpC, Fn1_β5–β6_ had a two-turned helix structure that was on the surface area, pointing away from the PR-domain. The other structure homologs had shorter Fn1_β5–β6_ linker regions. ScpA contained the cell adhesion motif RGD at the solvent exposed C-terminus of its Fn1 domain, whereas PrtS contained the potential inactive version of the cell adhesion motif RGE [16,38]. The Fn1 domain of ScpC ended with the sequence pattern KGQ, while the other Fn1 homologs lacked positive charges and instead had the C-terminal sequence pattern [YF]G[DQS].

The large Fn2 domains with 157–217 aa also showed structure homology when superimposed to PrtP_MS22337_ (RMSD < 3.2 Å) (Figure 6B). The sequence identity for the Fn2 domains of the different clusters was in the range 9–48%. The sandwich fold was surrounded by several loops and secondary structures that differed among the PrtP homologs. Fn2_β1–β2_ was a large and variable region in terms of both sequence length, loop formation, and secondary structure. This region of ScpA had previously been divided into two loop regions, where the first loop region of 733–757 was suggested to contribute to substrate binding [16]. Fn2_β4–β5_ orientated towards the active site with approximately 15 Å from the catalytic triad when Fn2_β4–β5_ formed an extended β-turn structure or a β-hairpin. This region had also been proposed to be involved in substrate specificity [21]. However, Fn2_β4–β5_ was much shorter in PrtP_NFICC96H_ than in the other PrtP homologs, locating Fn2_β4–β5_ further away from the catalytic cleft. Fn2_β5–β6_ pointed away from the active site and was more close to Fn3. This region formed a β-hairpin in most of the PrtP homologs, whereas Fn2_β5–β6_ of ScpA, PrtS, ScpC, PrtB, and PrtL had a shorter turn structure. In the phylogenetic tree, all Fn2 domains clustered together, but this cluster also included Fn4 of ScpC (Figure 5). When this Fn4 domain was superimposed on the Fn2 domains of PrtP_MS22337_, ScpC, and PrtP_NFICC96H_, the RMSD values were 3.2, 2.9, and 2.1 Å, respectively (Figure 6B). This suggested that the Fn4 domain of ScpC had a Fn2-like structural fold, but the loop regions Fn2_β1–β2_ and Fn2_β4–β5_ were shorter, indicating that the Fn4 domain of ScpC has another functionality.

The Fn3 domain was the smallest in the A-domain region, with 102–121 aa. This domain had two structurally different folds, both with fibronectin type-III-like domain cores (Figure 6C,D). The Fn3 domains of ScpA, PrtS, and ScpC made up a phylogenetic cluster together with Fn5 of ScpC (Figure 5). The domains of this cluster differed from the other Fn3 domains and had RMSD-values below 1.8 Å when superimposed on ScpC’s Fn3 (Figure 6D). The Fn3_β3–β4_ and Fn3_β5–β6_ regions had different structures in the homologs. ScpA had no β-hairpin in Fn3_β3–β4_, but was the only homolog with a β-hairpin in Fn3_β5–β6_. Despite the distribution of the Fn3 domains in the phylogenetic tree, the other Fn3 domains showed structure homology when superimposed on the Fn3 domain of PrtP_MS22337_ (RMSD < 1.9 Å) (Figure 6C). Fn3_β5–β6_ had a helical structure that pointed away from the PR-domain and towards the first Fn-domain in the B-domain region.

The phylogenetic tree of the Fn-domains suggested that the Fn-domains entailed more sequence variations in the B-domain region than in the A-domain region (Figure 5). The largest phylogenetic clusters of Fn4, Fn5, Fn6, and Fn7 each had a characteristic sandwich fold with similarities to a fibronectin type-III-like domain (Figure 6E–H). The two Fn10 domains did not adopt a well-defined structure in the AF models, but were predicted to be prone for β-strand structures by NetSurfP-3.0. The Fn8, Fn9, and Fn10 domains were in the same phylogenetic cluster, in which the Fn8 and Fn9 domains had the same structural fold with RMSD values below 2.2 Å when superimposed on Fn8 in PrtP_MS22337_ (Figure 6I). This suggested that the Fn8, Fn9, and Fn10 domains were repeating domains, probably the result of sequence duplications. A duplication event of the Fn2 and Fn3 domains of ScpC might also have generated the Fn4 and Fn5 domains, indicated by their sequence and structural similarities outlined above (Figure 6C,D). The Fn6 and Fn7 domains of ScpC shared sequence and structural similarities with the Fn4 and Fn5 domains in PrtS, respectively (Figure 5 and Figure 6J,K). These domains appeared to differ from the other Fn-domains. The Fn5 domain of PrtH2 and the Fn4 domains of the PrtP homologs of Cluster XII had sequence and structural similarities, but differed from the other Fn-domains (Figure 6L). PrtP homologs from Cluster XII had the same B-domain region as PrtH2, but lacked the Fn4 domain (Figure 2). As a result, the Fn5 and Fn6 domains of PrtR, PrtP_NFICC80_, and PrtP_NFICC200_ corresponded to the general Fn6 and Fn7 domains (Figure 6G,H). The B-domain region of PrtP_NFICC96H_ showed structural deviations, including its Fn8 domain that shared similarities with the general Fn5 domain. The other Fn-domains of PrtP_NFICC96H_ could not be properly classified as these structures were marked with too high uncertainties, as reflected with the low pLDDT (Appendix A).

In total, the Fn-domains of the PrtP homologs were categorized into 12 structurally different domains (Figure 6). The conserved Fn1 and Fn2 domains were characterized by their close spatial proximity to the PR-domains of the PrtP homologs. The Fn3 domains provided the transition to the B-domain region with different Fn-domain compositions, creating a backbone structure of repeating sandwich folds with five-to-seven β-strands. Hereby, some Fn-domains deviated from the strict definition of a fibronectin type-III-like domain, but the typical Fn-domains in the B-domain regions (Figure 6E–L) were around 4 nm measured parallel to their sandwich folds, just like the Fn3 domain. The sandwich fold of Fn1 was around 6 nm, while it was around 5 nm for Fn2. The backbone of the A-domain region would, in an extended rod-like structure, be around 15 nm, whereas the different B-domain regions could potentially contribute to further 8–28 nm.

#### 3.3.4. Helix Domain

PrtP homologs from Cluster II, IX, and X each had an H-domain following their B-domains, with nine, three, and eight helices, respectively (Figure 7A–C). PrtH (Cluster IX) had the smallest H-domain with 82 aa, forming a three-helix bundle. The H-domain of PrtS (Cluster II) had 209 aa and a stalk-like structure with three similar helical domains with a 32–38% sequence identity. Each domain consisted of a bundle with three helices and had a similar fold as the H-domain of PrtH (RMSD < 1.6 Å), to which they shared an 18–21% sequence identity. The H-domains from the PrtP homologs in Cluster X had approximately 220 aa with sequence identities ranging from 96 to 98%. These H-domains were made up of eight longer helices that were divided into two structural regions. The structural fold of the N-terminal region was similar to the H-domain in PrtH (RMSD around 2.8 Å), whereas the C-terminal region was larger, with five longer helices. In the structure models, the C-terminal region had two different orientations, indicating a potential flexibility of the H-domain (Figure 7D) with the possibility to adopt more compact structures, as previously predicted [21]. The structural similarities in the H-domains suggested that these H-domains had similar functions. Functional diversity might be encoded in the length of the H-domain that served as a potential spacer region to provide distance to the cell surface of the bacteria [5].

#### 3.3.5. Cell Wall Spacing Domain

The cell wall spacing domain, also known as the W-domain, preceded the cell wall attachment domain and ranged from 36 to 234 aa in length (Figure 2). All PrtP homologs with cell wall attachment domains contained a W-domain, whereas the W-domain was absent in PrtP_NFICC6H_, as was the cell wall attachment domain.

The W-domains appeared as protein regions with no well-defined globular structures in the AF models. Instead, the W-domains of the AF models had very low pLDDT, indicating that these regions were characterized by ID (Appendix A). NetSurfP-3.0 corroborated that the W-domains were characterized by ID, which distinguished the W-domain from the neighboring domains. The W-domains were quite hydrophilic, with a generally low content of hydrophobic residues and an almost complete lack of Cys, Met, and aromatic residues (Phe, Tyr, and Trp). Low complexity characterized the sequences as these were rich in certain aa residues including Ala, structure-breaking residues (Pro and Gly), charged residues (Asp and Lys), and polar residues (Ser, Thr, Asn, and Gln). However, the dominating aa residues of the W-domains differed, resulting in acidic (theoretical pI < 5.5) and basic (theoretical pI > 9.0) W-domains. The sequence identity was below 19% for the PrtP homologs across the different clusters. Sequence diversity of the W-domains also appeared within clusters, particularly Cluster X and XII, that had a 33–93% and 6–51% sequence identity, respectively. The W-domains had an 87% and 65% sequence identity in Cluster V and VI, respectively. Hereby, the W-domain appeared as the most variable domain in the PrtP homologs.

In Cluster X, the W-domains of the PrtPs ranged from 69 to 189 aa, with an imperfect modular repeat unit of 60 aa. As previously reported [21], this unit was repeated up to three times and appeared to be unique to this group of PrtPs, as the same tandem repeat was not observed in other W-domains. Two other long tandem repeats were located in the W-domains of PrtH2 and ScpA using T-REKS and XSTREAM. PrtH2 had the longest W-domain, with a tandem repeat of 39 aa that was repeated 2.3 times. The sequence of ScpA repeated 17 aa 3.7 times, as previously reported by Siezen [5]. The repeats were located at the N-terminal in the W-domains of PrtH2 and ScpA, whereas the unit of 60 aa made up all the W-domains of the PrtP homologs in Cluster X (Appendix A). The W-domain with this sequence unit of 60 aa facilitated cell adhesion between *Lactococcus* cells as well as between *Lactococcus* and epithelial cells [39]. Interestingly, the number of tandem repeats increased the degree of cell interaction, which was driven by protein–protein interactions. The W-domain with a single sequence unit was less accessible for interactions that hindered or disrupted cell adhesion when the PrtP was cell anchored [39]. In this case, almost 60 residues appeared to be necessary for displaying a functional protein domain on the bacterial surface, though more than 90 aa was needed in other cases [40].

Except for the W-domains of PrtP_SK11_, PrtP_MS22337_, and PrtP_MS22333_, the W-domains preceding an LPXTG-like motif ranged from 69 to 94 aa, which in a fully expanded form could span a cell wall of 19–26 nm. The length of these W-domains might affect the efficiency of protein anchoring and protein stability, as described for other Gly/Pro- and Ser/Thr-rich low complexity linker regions preceding LPXTG-like motives [41]. The W-domains of PrtH, PrtB, PrtL, and PrtH2 preceded a SlpA-domain, suggesting another functionality of these W-domains than to be cell wall spacers. These W-domains had large sequence variability ranging from 45 to 234 aa and shared no obvious sequence characteristics such as charge or sequence patterns. Hereby, the ID characterized W-domains appeared to have at least a dual role in PrtP homologs facilitating surface exposure and/or mediating interactions for cell-wall-attached PrtP homologs.

#### 3.3.6. Cell Wall Attachment Domains

The cell wall attachment domains of the PrtP homologs followed the W-domain towards the C-termini (Figure 2). The presence of either an AN-domain or a SlpA-domain reflected two distinct cell envelope attachment mechanisms. Interestingly, *Leuconostoc* PrtP_NFICC96H_ lacked both of these attachment domains and instead terminated after a tail of eight fibronectin-like domains. The lack of a cell wall attachment domain had also been observed for *Lactobacillus* PrtPs [6], implying that termination without an attachment domain might be a widespread domain architecture of PrtP homologs.

Each of the AN-domains started with an LPXTG-like motif, then a region with hydrophobic aa and a short tail of charged residues (Figure 8). This domain structure corresponded to other LPXTG anchoring domains recognized by the transpeptidase enzyme sortase [42,43]. Sortase mediates the covalent attachment of secreted proteins to cell walls by cleaving the canonical LPXTG motif between Thr and Gly. This canonical motif was found in six of the PrtP homologs, while the putative motives for PrtP_NFICC96Q_, PrtP_NFICC96W_, Prt_NFICC120_, PrtR, and ScpA were LAKTA, LPDTA, FPTTN, MPQAG, and LPTTN, respectively. The motives of PrtR and ScpA had previously been reported, but without experimental verification [14,16]. However, functional variants of the canonical motif existed among Gram-positive bacteria, which entailed a tendency for species–specific motif patterns [6,43]. Sortase variants had also displayed altered substrate specificities, allowing substitution within the motif, such as L→M, P→A, and T→[ALSV] [42,44,45]. These motif variations were consistent with the observed motif patterns in the PrtP homologs, indicating that the AN-domains, along with the conserved C-terminal region, were most likely functional attachment domains.

The SlpA-domains occurred as tandem pairs of the Pfam domain PF03217, forming a bipartite three-dimensional structure (Figure 9). Surface layer proteins such as SlpA form the functional crystalline monolayers of some bacteria, including several but not all *Lactobacillus* species [46]. The SlpA exists in different proteins, in which they can mediate non-covalent attachment to the bacterial cell wall. Among the analyzed PrtP homologs, the SlpA-domains were only represented in the *Lactobacillus* PrtP homologs PrtH, PrtH2, PrtB, and PrtL (Figure 2). These SlpA-domains contained 114–136 aa, where the repeating regions consisted of approximately 60 aa and shared sequence identities ranging from 11 to 95%. All four SlpA-domains were rich in Lys residues (17–32%) and depleted of acidic aa residues, resulting in domains with high theoretical pI values (pI > 10). As a result, the SlpA-domains would be protonated within the pH range for the optimal activity of PrtP homologs, complementing the negative charges of the bacterial cell wall. Electrostatic forces were the main contributor for PrtL attachment [12], supporting the cell wall anchoring function of the SlpA-domain within PrtP homologs. The AF structure models of the SlpA-domains were predicted with diverse qualities, which did not support a homology assessment of their predicted three-dimensional structures. The structures of the SlpA-domains of PrtL and PrtH2 were uncertain, reflecting unlikely structures that could not be divided into two separated structural domains. On the other hand, the structural models of the SlpA-domains of PrtB and PrtH had pLDDT generally above 70, suggesting a generally good backbone prediction (Appendix A). The overall structures of these SlpA-domains had a similar three-dimensional dumbbell shape, organizing the β-sheets in two regions (Figure 9). This structural fold corresponded to the structure of other SlpA-domains, where the two β-sheet regions were divided by a spacer region with a reduced sequence identity compared to the two tandem regions [46,47].

## 4. Discussion

### 4.1. Conserved Domains in Novel Domain Architectures

AF modeling has provided structures of diverse PrtP homologs, representing different LAB species from dairy, plant, and human ecological niches. These AF structural models, together with published crystal structures of ScpA and ScpC, have facilitated novel insights into the multi-domain architectures of PrtP homologs. Additionally, a phylogenetic analysis groups the PrtP homologs into 12 clusters, of which PrtP homologs of the same cluster share domain architectures.

Domain boundaries have been specified, showing that PrtP homologs can have up to 16 different domains. The domains constitute 10 overall domain architectures of the analyzed PrtP homologs, in addition to the simplest domain architecture of subtilisin. These overall domain architectures of the PrtP homologs differ from each other by the presence or absence of the domains PA, H, AN, and SlpA, and by the number of Fn-domains in the B-domain region. For each of the 12 phylogenetic clusters, the PrtP homologs share the same domain architectures. Cluster V, VII, and VIII share the same overall domain architecture, whereas each of the remaining nine clusters have different domain architectures. This general pattern suggests that the function of the catalytic region is evolutionarily connected to its flanking regions, particularly the longer and divergent C-terminus. The specified domain boundaries increase the resolution for comparative studies among PrtP homologs as single domains can be compared instead of regions composed by different domains.

The division of the A- and B-domain regions into Fn-domains reveals conserved patterns, emphasizing that the Fn-domains have important and different functionalities. PrtP homologs can have up to 10 Fn-domains in different architectures that are composed by 12 structural groups of Fn-domains. Fn1 and Fn2 are the most conserved Fn-domains along the tails of fibronectin-like domains. Gene duplication events may have facilitated the extension of fibronectin-like tails, and these events may have evolved from the A-domain region as suggested by the potential duplication of the Fn2 and Fn3 domain regions in ScpC. All PrtP homologs of LAB have at least three conserved Fn-domains, corresponding to the A-domain region, whereas the different lengths of the B-domain regions correspond to the number of its Fn-domains. Therefore, the bipartite division of the tail of Fn-domains into two regions is still appropriate. In this paper, we refer to these regions as the A-domain region and the B-domain region, based on the previously defined A-domain and B-domain of PrtP homologs.

Different cell-envelope attachment strategies contribute to the number of domain architectures among the PrtP homologs. The comparative structural analysis specifies that the W-domain and the SlpA domain should be defined as two separate domains. Cell wall attachment by surface layer proteins such as the SlpA-domain has previously been suggested for PrtB, PrtL, PrtH, and paralogs to PrtH2 [2,5,12]. The W-domain is characterized as an ID region by bioinformatics analysis, which corresponds to the ID characterization of the W-domain in ScpC by biophysical analyses [20]. In contrast, the SlpA attachment domain appears to adopt a well-defined three-dimensional structure in the AF models. The structural difference between the W-domain and the SlpA-domain indicates that these domains have potentially different functionality, emphasizing that clear domain definitions are essential. The delimitation of the SlpA-domain clarifies that PrtP homologs used two different strategies for cell envelope attachment. The comparative structural analysis supports that cell envelope-attached PrtP homologs have either a SlpA-domain or an AN domain and not both. The AN domain is not only determined by the presence of the canonical LPXTG motif or a variant of it, but requires a C-terminal characteristic region of hydrophobic residues followed by charged residues. Among the analyzed PrtP homologs, the SlpA-domains have only been identified in the *Lactobacillus* PrtP homologs, though SlpA is not restricted to the *Lactobacillus* species, as observed in pfam. The presence of the AN domain cannot be directly translated to PrtP attachments, depending on both sortase activity and environmental conditions [12,15,48]. Other PrtP homologs lack an attachment domain. PrtP homologs with no cell wall attachment domains may have greater industrial importance as secreted proteases, which can be obtained by selecting or bioengineering PrtP homologs.

The ID-characterized W-domains have the highest sequence heterogeneity among the domains of the PrtP homologs. Therefore, the W-domains may seem to be less conserved compared to the other domains. On the other hand, protein ID regions can disappear during evolutionary time, where random aa substitutions can readily drive the conversion of ID regions into structural regions [49]. Hereby, the W-domains are likely to have evolutionary conserved functions as ID regions need to be maintained by evolutionary constraints. ID regions are associated with conformational flexibility and are great mediators for protein interactions, which correspond to the suggested function for the W-domains. The W-domains might have different functions among the PrtP homologs, as suggested by their different pI-values and tandem repeats, which may bring more novel domain architectures to the PrtP homologs. However, this present analysis could not find a basis for the further division of the W-domains.

### 4.2. Conformational Maturation and Stability

The catalytic activities of both ScpA and subtilisin rely on the assistance of their propeptides. ScpA is active when expressed with its ID-characterized propeptide, but not without this propeptide [16]. The structured propeptide of subtilisin has been proposed to stabilize the core structure of PR in order to achieve proper folding [50]. Hereby, the ID and structurally characterized propeptides are intramolecular chaperones in the PrtP homologs, where the propeptides are required for proper folding and catalytic activity. The structural propeptides with the prolonged α-helical structure have a larger interface with the PR-domain than the interface between the propeptide and PR-domain in subtilisin. The larger interfaces include the interaction to a short helix of the PR-domains, which is absent in the smaller PR-domain of subtilisin. This interaction may also assist the folding process of the PR-domains.

The propeptide is not the only required factor for assisting the maturation process of PrtP homologs, some of which require the prolyl *cis*/*trans* isomerase activity of the protein folding chaperone PrtM to achieve their active conformations [51]. The proline residues required for these proteases’ conformational changes have not been identified, but critical proline residues have been proposed to be located in the PR-domains as the autocatalytic processing can depend on the absence of PrtM [21,51]. However, the PrtP homologs may follow different maturation steps, as not all PrtP homologs seem to depend on PrtM maturation [8,13,15], and ScpC is cleaved in its PR1, working as a heterodimer (Figure 2) [17,20]. The different maturation processes of the proteases are not included in the structures proposed by the AF modeling.

The structure models of the PrtP homologs arrange the multiple domains in different structural conformations where the Fn-domains of the B-domain regions coil back on the structure. These conformations show possible flexibility in the PrtP homologous structures, in which domains along the C-termini can approach PR, including the catalytic cleft. The crystal structure model of ScpC shows an interaction between the PR-domain and the two C-terminal domains Fn6 and Fn7 [17]. This bended conformation has been suggested to be stabilized by interfacial calcium ions between the Fn-domains from Fn2 to Fn5 [20]. The crystal structure of ScpA contains a single calcium ion that is bound to Fn2 [16]. Changes in environmental calcium ion levels have been proposed to induce conformational changes in PrtP homologs. The coiled conformation of ScpC has been suggested to be more extended in the absence of calcium ions [20], whereas the conformation of PrtP_MS22337_ has been proposed to be extended, stabilizing the rod of Fn-domains that keeps the catalytic domain away from the cell envelope in the presence of calcium ions [21]. A similar calcium-dependent strategy is discovered in bacterial adhesins, which also contain tandem β-sandwich domains [52]. The linker regions between the β-sandwich domains of the adhesins are coordinated by the presence of calcium ions, facilitating a rigidification of their rod-like structures. The depletion of calcium ions may facilitate the switch to a more bended conformation of PrtP_MS22337_, in which the Fn-domains of the B-domain region can approach the catalytic domain, as proposed by the AF structure.

Such bended conformation may explain the autoprocessing of the C-termini in the PrtP homologs from Cluster X, by which these proteases are released from the cell envelope in media depleted for calcium ions. However, smaller conformational changes within a bended conformation may also explain the self-release. The autoprocessing cleavage site for the release has not been determined, but it is proposed to be in the B-domain region near the aa position 1459 in PrtP_SK11_ [53], suggesting a cleavage site in the flexible linker region between Fn6 and Fn7 or between Fn7 and Fn8. These C-terminal autoprocessing sites will lead to a different stability of the proteases as instability increases with C-terminal truncations of the B-domain region [53]. After autoprocessing, calcium ions can restore the stability and activity of PrtP_Wg2_, whereas PrtP_SK11_ only partly restores its stability and activity [48]. This difference may rely on their different enzymatic activity and substrate specificity [54], but different autoprocessing cleavage sites within PrtP_Wg2_ and PrtP_SK11_ may also explain differences in the stability of the C-terminal truncated proteases.

The deletion of the whole PA-domain has no influence on the autoprocessing of PrtP_SK11_ [55] and proteases without a PA-domain can also undergo autoprocessing such as PrtR [14]. PrtR has a shorter B-domain region with only three Fn-domains that structurally differ from the Fn-domains of the PrtP homologs from Cluster X. On the other hand, the depletion of calcium ions does not release all PrtP homologs such as PrtL [2,12], though it has a long B-domain region of seven Fn-domains. The B-domain region may obtain the stability of the proteases by its composition of Fn-domains rather than the presence or absence of single Fn-domains, though other domains such as the catalytic domain can affect the stability. The determination of the structural different Fn-domains may help to elucidate their roles within different domain compositions.

### 4.3. Substrate Selectivity and Specificity

Even though some part of the B-domain region may approach the PR-domain, the B-domain region and its C-terminal flanking domains are not determinants for substrate specificity [53]. The *Lactococcus* proteases PrtP_Wg2_ and PrtP_SK11_ from Cluster X display different substrate specificity, though only 45 aa differ along the first 1800 aa positions. More than half of these aa positions are located in the catalytic domain and A-domain regions. The PR1 regions of PrtP_Wg2_ and PrtP_SK11_ differ at eight aa positions, including the substrate binding region defined by Exterkate et al., 1993 [9], whereas PA and Fn2 differ at 11 and 5 aa positions, respectively. These sequence variations suggest that PA and Fn2, together with PR, are involved in the complex modulation of substrate specificity among PrtP homologs [10]. The structure models of the PrtP homologs show spatial proximity between the catalytic cleft and both the PA and Fn2 domains, supporting that both domains can modulate enzymatic activity together with PR.

The deletion of the PA-domain changes the substrate specificity of PrtP_SK11_ and reduces its caseinolytic activity [55]. The caseinolytic activity of PrtR is modest, which may be related to the absence of a native PA-domain [14]. On the other hand, subtilisin is without a native PA-domain characteristic by its broad substrate specificity and its exquisite catalytic efficiency. The PrtP structure models of Cluster I–X show the structure homology of their PA-domains that appear as structural units in the catalytic domain regions. A characteristic antiparallel β-strand structure is conserved and indicates the transition between the PR and PA-domains. This transitional β-sheet structure may permit the removal of PA without introducing large conformational changes in PR, as previously suggested by the model structures of PrtP_MS22337_ with and without its native PA-domain [21]. The transitional β-sheet structure appears to be a conserved feature, which may contribute to the flexibility of the PA-domains. PA-domains show conformational mobility within the biophysical structure analysis of ScpC and other SpyCEPs [20]. This conformational flexibility may modulate the accessibility to the catalytic sites of the proteases, which have been proposed to be regulated by an allosteric link between the catalytic site and the PA-domain.

No evidence exists to date for the dimerization of cell envelope PrtP homologs in bacteria, whereas the PA-domain of a pyrolysin-like subtilase from tomato plants performs an essential role in both the homo-dimerization and regulation of proteolytic activity [56]. Different aa positions within their PA-domains have been suggested to play a role in substrate specificity, though they point away from the catalytic cleft and binding region [10]. The majority of the 11 different aa positions between PrtP_Wg2_ and PrtP_SK11_ are on the same surface area that points towards the catalytic cleft and substrate binding region, indicating their potential regulatory role in modulating enzymatic activity. However, these aa positions have not been identified as determinants for substrate specificity among different groups of *Lactococcus* PrtP homologs [10]. However, the PA-domain appears to assist substrate binding in ScpA, mediating the communication between the substrate-binding region and the active site [16,18]. In this way, the conserved PA-domain structure may regulate accessibility to the catalytic site, ensuring the proper substrate selectivity and specificity of bacterial PrtP homologs with multiple domains.

The PrtP homologs appear to manage substrate binding by regions in primarily PR1 and Fn2. The caseinolytic specificity of PrtP_Wg2_ and PrtP_SK11_ has been modified in hybrid proteases, of which regions of PR1 and Fn2 have been swapped between these homologs [9,57]. Five of the eight different aa positions within PR1 have been identified as being determinants for the substrate specificity of PrtP homologs from Cluster X. These aa positions correspond to the substrate-binding region of subtilisin, suggesting that this region of PrtP homologs in general plays a role in substrate-binding. In contrast to subtilisin, PrtP homologs with multiple domains contain each a conserved Fn2 domain, which plays an essential role in the substrate binding and mediation of substrate specificity. The phylogenetic clustering of the Fn-domain sequences suggests that the Fn2 domains have conserved and unique characteristics among the Fn-domains. The Fn2 domain is the largest domain in the tail of fibronectin-like domains, and its model structures show interesting variable loop regions.

Fn1 connects PR and Fn2 and keeps them in close contact, where loop regions of Fn2 can point towards the catalytic cleft. Fn2_β4–β5_ is among these loop regions closest to the catalytic cleft. The C-terminal β-hairpin or corresponding β-turn structure in Fn2_β4–β5_ (PrtP_MS22337_; residues 936–945) is located approximately 15 Å away from the catalytic cleft, whereas PrtP_NFICC096H_ is the only homolog that lacks this C-terminal loop structure. The N-terminal loop region is shared among all the PrtP homologs and corresponds to the region (PrtP_MS22337_; residues 929–935) that is known to play a role in the substrate specificity of PrtP homologs from Cluster X [9,57]. These N-terminal regions of PrtP_Wg2_ and PrtP_SK11_ differ in two positions, and the swap of these aa residues results in the hybrid specificities of these proteases [57]. This supports the idea that this loop region is one of, but not the only determinant in the substrate specificity of these PrtP homologs, whereas the role of the hairpin and turn structures is unknown. Fn2_β1–β2_ is another loop region that points towards the catalytic cleft, and it has been suggested to entail a substrate-binding region in ScpA [16]. This binding region of Fn2_β1–β2_ shows structural variations among the homologs that may be related to their different substrate preferences. The major contribution to the substrate binding energy has recently been assigned to the Fn2 domain of ScpA, which recruits its substrate by long-ranged electrostatic interactions [18]. Complementary electrostatic interactions have also been proposed to be involved in the substrate interaction for other PrtP homologs [9,58], suggesting that the substrate binding mechanisms may be similar among different PrtP homologs.

The interplay between the domains including PR, PA, and Fn2 seems to regulate the enzymatic activity of the PrtP homologs rather than single determining factors. The variable loop regions stand out as potential determining binding areas for substrate selectivity and specificity, whereas the core domain structures are conserved, ensuring the required conformational stability and flexibility for governing substrate selectivity and specificity.

### 4.4. Adhesion Potential

The PrtP homologs contain different domains with adhesion potentials, indicating that PrtP homologs may mediate adhesion. PrtP homologs have been shown to have adhesive properties against polystyrene, mucin, and fibronectin [59,60]. The proteolytic activity is not responsible for the adhesion [59], but adhesion to epithelial surfaces has been reported to rely on the cell adhesive motif pattern RGD in ScpA and the W-domain in PrtP_SK11_ [39,61]. The RGD motif and variations of this motif pattern are located in a few of the PrtP homologs within the furthest C-terminal end of Fn1. An additional RGD motif has been reported in the PR-domain of ScpA [61]. However, all the cell-wall-attached PrtP homologs contain quite diverse W-domain regions with under-investigated adhesive potentials. The W-domain of PrtR shows a high sequence identity with the C-termini regions of two cell surface antigen I and II polypeptides that provide adhesion to glycoproteins [14]. The structure family of the Fn-domains belongs to the overall immunoglobulin-like (Ig-like) beta sandwich fold (SCOP ID: 2000051), which is present in some adhesins. Therefore, the Fn-domains may also have adhesive properties though the adhesion against epithelial cells is not connected to Fn-domain regions of PrtP_SK11_ [39]. A model of the interaction between PrtP_MS22337_ and a casein micelle has been proposed, relying on the adhesive properties of the protease [21]. The adhesion of PrtP homologs to milk proteins has not directly been shown, but dairy-isolated LAB strains show a stronger binding ability against milk proteins compared to LAB strains from plant isolates [62]. The diversity of W-domains and Fn-domains has an interesting adhesive potential that may play a selective advantage of LAB strains in different environments.

### 4.5. Distribution and Environmental Constraints

PrtP homologs are found in many different LAB species representing almost all LAB families, including *Carnobacteriaceae*, *Enterococcaceae*, *Lactobacillaceae*, *Leuconostocaceae*, and *Streptococcaceae*. In this present study, the PrtP homologs from the plant-derived bacteria are only theoretical proteins as their identification is based on genome data. The expression and activity of these proteases need to be validated to support that the PrtP homologs are present and biologically relevant in the LAB families *Carnobacteriaceae*, *Enterococcaceae*, and *Leuconostocaceae*. PrtP homologs are far from prevalent in all LAB strains [6,7], and plasmids encoding PrtP homologs can radially be lost during the propagation of LAB strains [63]. Therefore, the presence of genes encoding PrtP homologous protein sequences indicates that these proteases are biologically active.

The PrtP homologs from plant-derived LAB strains are classified into six clusters (IV, V, VII, VIII, X, and XII), with Cluster V representing LAB strains from five different plant materials: berry, evergreen, flower, fruit, seashore plant, and tap/tuber root. In general, the clusters of PrtP homologs are from LAB strains isolated from various environments, indicating that certain domain architectures are not correlated with the environmental origin of the LAB strains. Cluster XII contains PrtP homologs from *Leuconostoc lactis, Leuconostoc mesenteroides*, and *Lacticaseibacillus rhamnosus* strains isolated from berry, seashore plant, and human vaginal materials. PrtP_NFICC80_, PrtP_NFICC200_, and PrtR represent the diversity of Cluster XII, demonstrating that structurally similar proteases not only span different LAB species, but also span various natural LAB habitats.

The same LAB strain can contain structurally diverse PrtP homologs, as observed for the novel plant-derived *Leuconostoc mesenteroides* strain NFICC96 and several *L. helveticus* strains. The four PrtP homologs of the *L. helveticus* strain CNRZ32 are divided into two phylogenetic clusters with PrtH in Cluster IX and the other homologs PrtH2, PrtH3, and PrtH4 in Cluster XI, corresponding to the previous subdivision of their PrtP encoding genes [64]. Phylogenetic gene analysis suggests that the gene acquisition of *L. helveticus* has arisen by both gene duplication and horizontal gene transfer events [64]. However, the acquisition of multiple PrtP homologous encoding genes may not be reserved for the *L. helveticus* strains, as the plant-derived *Leuconostoc mesenteroides* strain NFICC96 potentially contains three PrtP homologs.

The similarity of the PrtP homologs among LAB isolates from different species and environments suggests that PrtP-encoded genes are shared among LAB strains via horizontal gene transfer. Hereby, the PrtP homologs may facilitate cross-over fermentations, in which a microbe of a traditional fermentation process is introduced to another protein substrate. This is supported by the introduction of the dairy-associated *S. thermophilus* strain of soy milk, in which PrtS_LMD-9_ facilitates bacterial growth by a modest degradation of soy proteins [65]. Extracellular proteases of LAB strains may play a central role in the development of plant-based foods as they have done in dairy products [24]. The activity of PrtP homologs may be predicted and steered, using the clarified domain map for domain swapping.

## 5. Conclusions

Structure modeling has been performed using AF, which has created a platform for the structural comparative analysis of PrtP homologs from diverse LAB strains. The delimitations of the protein domains have revealed more detailed domain architectures of PrtP homologs. The phylogenetic clusters of PrtP homologs have different domain architectures that do not necessarily correlate with the natural habitat or species of the bacterial strains. Horizontal gene transfer may explain the occurrence of similar PrtP homologs in different LAB species that are isolated from different ecological niches. The core folds of the structural domains are often evolutionarily conserved among the PrtP homologs, whereas variations are addressed to different loop regions. The PrtP homologs contain different combinations of fibronectin type-III-like domains. The Fn-domains could be divided into 12 structural groups, of which Fn1 and Fn2 would show the highest evolutionary conservation. Fn2 plays a role in substrate interaction together with other domains of the proteases. The functions of the other Fn-domains are more speculative, but are likely to contribute to the conformational stabilization of the PrtP homologs. The structure analysis of the domain boundaries has also emphasized the presence of an ID-characterized W-domain, preceding two different anchoring domains. The variable W-domains may mediate different protein interactions. Substrate interactions of PrtP homologs are complex, regulated by interactions between different domains. Domain–domain interactions may also affect the conformational changes of the PrtP homologs, with potentially different responses on environmental changes. This highlights the importance of defined domain boundaries in studying the impacts of different domains and domain combinations on the stability, proteolytic activity, and potential adhesive activity of PrtP homologs.

## Figures and Tables

**Figure 1 microorganisms-11-02256-f001:**
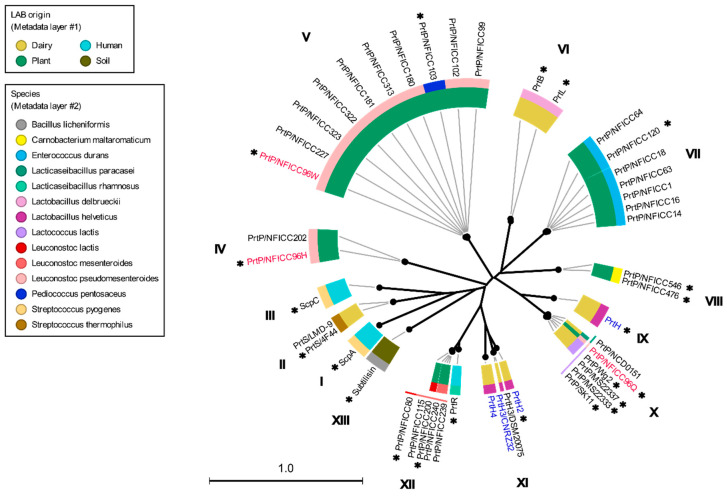
Phylogenetic tree and clusters of PrtP homologs. The radial phylogenetic tree was based on the analysis of protease domain regions of 45 PrtP homologs, including protein sequences corresponding to the protease (PR) domain region with matching protease-associated (PA)-domains. The names of the LAB strains were provided as part of the protease name to distinguish between the proteases with the same name. The LAB strains derived from plant sources were from the National Food Institute Culture Collection (NFICC). The neighbor-joining method was used to generate the tree topography, using the CLC Main Workbench [25]. Black branch lengths were proportional to the Jukes–Cantor distances, indicated by the scale bar. Bootstrapping with 100 replicates was used to assess the tree reliability. All branches were verified with bootstrapping above 70%, as shown by the thick lines. PrtP homologs were found in bacteria strains of different species (outer circle layer #2) that were derived from different origins (inner circle layer #1). Three PrtP homologs (red) were from *Leuconostoc pseudomesenteroides* NFICC96 and the four PrtP homologs (blue) were from *Lactobacillus helveticus* CNRZ32. The numbers I–XIII were assigned to the clusters of PrtP homologs, counting clockwise after subtilisin. PrtP homologs marked with asterisks were included for further structure analysis.

**Figure 2 microorganisms-11-02256-f002:**
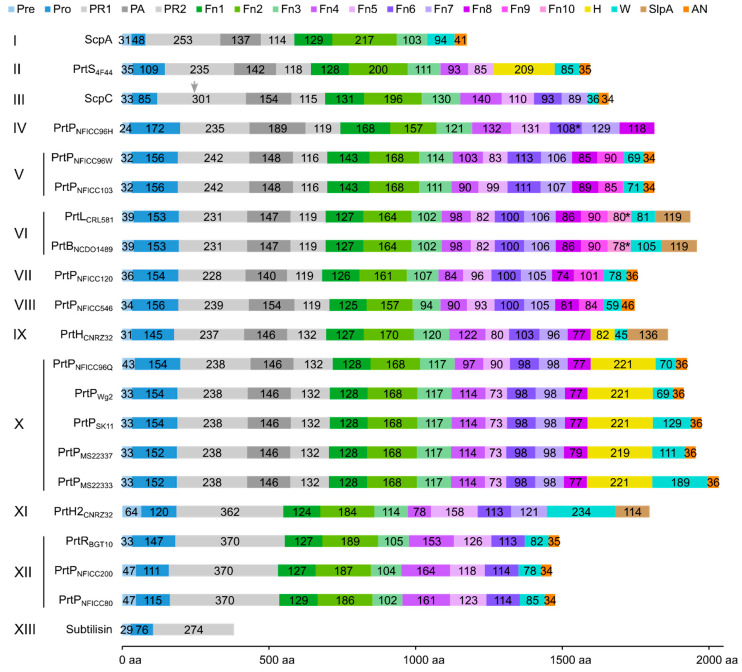
Protein domain architectures of PrtP homologs. The domain architectures of PrtP homologs were depicted as bar plots with colored boxes representing domains. The numbers within the boxes referred to the domain size of amino acids (aa), whereas the domain position was given on the horizontal axis. Based on sequence similarities in the catalytic domain regions, the PrtP homologs were classified into 13 phylogenetic clusters (I–XIII) (Figure 1). PrtP homologs from the same cluster, as well as PrtP homologs from cluster V, VII, and VIII, had similar domain architectures. As a result, the PrtP homologs represented 10 different domain architectures. The pre-propeptide (PP) domain (blue colors) was made up of the prepeptide (Pre) with a secretory signal and the propeptide (Pro). The catalytic domain region (gray colors) contained the protease (PR) domain, which was divided into two parts (PR1 and PR2) by the protease-associated (PA)-domain. The A-domain region (green) and B-domain region (purple) were constituted by the fibronectin-like III (Fn) domains numbered 1–10 from the N-termini. The helix (H) domain, cell wall spanning (W) domain, cell wall anchor (AN) domain, and surface layer protein A (SlpA) domain were additional domains. The domain delimitations were based on the analysis of three-dimensional protein structures with supplementary bioinformatics structure analyses. If the domain had not been assembled as a domain unit in the AlphaFold 2 structure models, it was indicated by a star (*). ScpC existed as a heterodimer (PDB ID: 5XYR) after maturation, in which ScpC was cleaved in PR1 between amino acid 244 and 245 [17], as indicated by a gray arrow.

**Figure 3 microorganisms-11-02256-f003:**
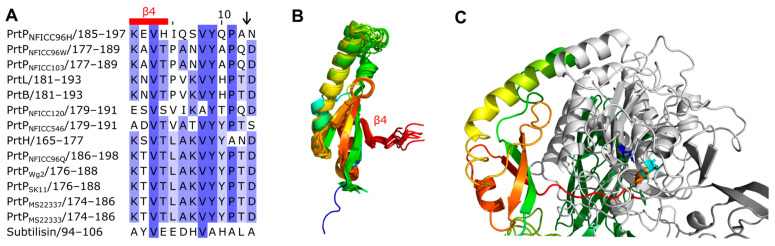
Cleavage-site and fold of structural propeptides. (**A**) Protein sequence alignment showed the cleavage region in PrtP homologs with structural propeptides. The amino acid (aa) residues were colored blue with increased sequence identities. The region included the four aa long β4 strand and the cleavage site (↓). (**B**) The propeptides were colored from the N-termini (blue) to the C-termini (red), but the N-terminal helices of the propeptides were not depicted. The structures were modeled by AlphaFold 2 (AF) based on the whole protein sequence of the PrtP homologs in the alignment. The propeptide with the conserved structural fold β1-α1-β2-β3-α2-β4 was shown superimposed on the propeptide of PrtP_MS22337_ with RMSD < 2.1 Å. The RMSD-values were lower than 1.0 Å when the propeptide of subtilisin was excluded. (**C**) The complex between the propeptide and protease domain (light gray) was depicted for the AF structure model of PrtH. The catalytic residues Asp (blue), His (cyan), and Ser (orange) were highlighted as sticks.

**Figure 4 microorganisms-11-02256-f004:**
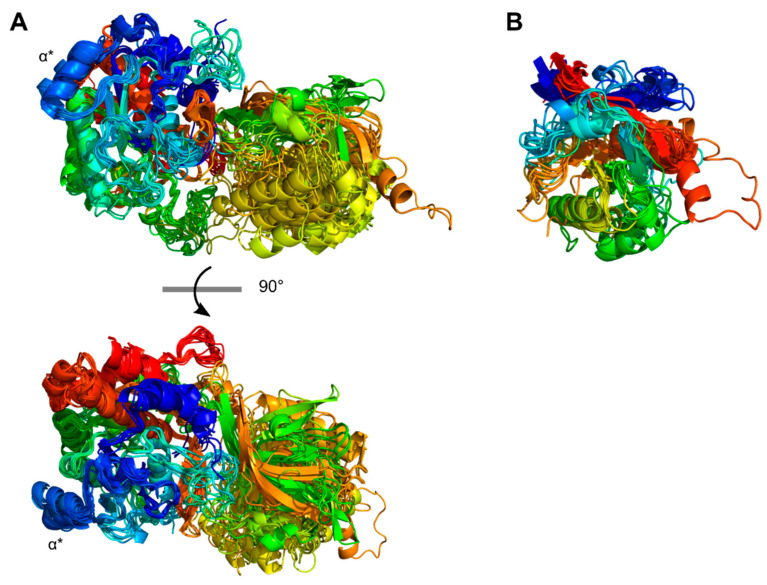
The catalytic domain region. (**A**) The structures of the catalytic regions were modeled using AlphaFold 2, including 21 PrtP homologs. The protease (PR) domain was located to the left in the protein structure. A protease-associated (PA)-domain was located in the catalytic domain regions of 16 PrtP homologs. The catalytic domain region was colored from the N-termini (blue) to the C-termini (red), giving the PA-domain green, yellow, and orange colors in the right part of the protein structure. The PR-domain structures were superimposed on the PR-domain of PrtP_MS22337_ (RMSD < 1.2 Å) and were shown from two angles. Superimposed structures of the entire catalytic domain regions gave 0.16–4.0 Å and were not displayed. The PrtP homologs had the S8 subtilase family PR domain alpha/beta fold, which included a parallel β-sheet with seven β-sheets. Except for subtilisin, all PrtP homologs had the three-turned helix (α*). (**B**) The protease-associated (PA)-domain was an insert domain between the two PR-domain regions PR1 and PR2. PrtH2, PrtR, PrtP_NFICC80_, PrtP_NFICC200_, and subtilisin did not have a PA-domain. The other 16 PA-domains were superimposed on the PA-domain of PrtP_MS22337_ that had a general fold β1-β2-β3-α1-β4-β5-α2-β6. The PA-domains were colored from their N-termini (blue) to the C-termini (red).

**Figure 5 microorganisms-11-02256-f005:**
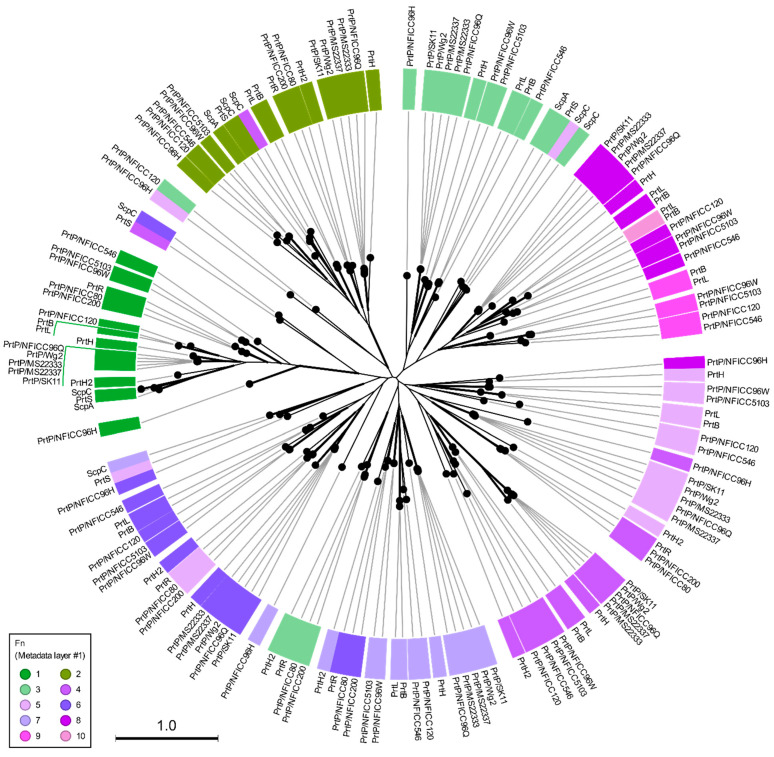
Phylogenetic clustering of fibronectin type-III-like (Fn) domains in the A- and B-domain regions. The radial phylogenetic tree was created to show phylogenetic clustering of all Fn-domains that were identified in 20 PrtP homologs. The PrtP homologs had up to ten Fn-domains creating a tail of Fn-domains that were numbered from the N-termini (metadata layer #1). The A-domain region of each PrtP homolog had three Fn-domain (green colors), whereas the B-domain region had up to seven Fn-domains (blue/purple colors). The neighbor-joining method was used to generate the tree topography. Black branch lengths were proportional to the Jukes–Cantor distances, indicated by the scale bar. Bootstrapping with 100 replicates was used to assess the tree reliability. Only highlighted branches were supported with bootstrapping above 70%. The names of the lactic acid bacteria (LAB) strains were provided as part of the protease name to distinguish between the proteases with the same name. LAB strains derived from plant sources were from the National Food Institute Culture Collection (NFICC).

**Figure 6 microorganisms-11-02256-f006:**
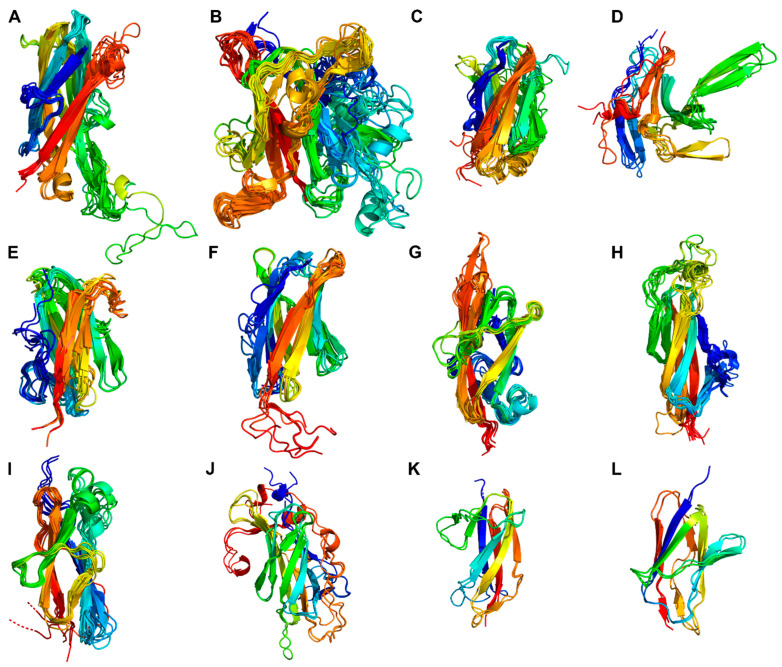
Structure homology of the fibronectin type-III-like (Fn) domains. The structures of the homologous Fn-domains were superimposed, showing 12 structurally different Fn-domains of 20 PrtP homologs from lactic acid bacteria. The Fn-domains were numbered 1–10 based on their positions within their respective PrtP homologs (Figure 2). Four Fn-domains (**A**–**D**) were observed in the A-domain region: (**A**) Fn1 domains, (**B**) Fn2 domains and the Fn4 domain of ScpC, (**C**) most Fn3 domains except for (**D**) the Fn3 domains of ScpA, PrtS, and ScpC together with the Fn5 domain of ScpC. Eight Fn-domains (**E**–**L**) were in the B-domain region. The B-domain region included the general Fn-domain structures: (**E**) Fn4 domains, (**F**) Fn5 domains, (**G**) Fn6 domains, (**H**) Fn7 domains as well as (**I**) the Fn8 and Fn9 domains that were superimposed together. A few Fn-domains deviated from the general observed Fn-domain structures, including: (**J**) Fn5 of PrtH2 and the Fn4 domains in PrtR, PrtP_NFICC80_, and PrtP_NFICC200_, (**K**) Fn4 in PrtS and Fn6 in ScpC, and (**L**) Fn5 in PrtS and Fn6 in ScpC. The domains were colored from the N-termini (blue) to the C-termini (red).

**Figure 7 microorganisms-11-02256-f007:**
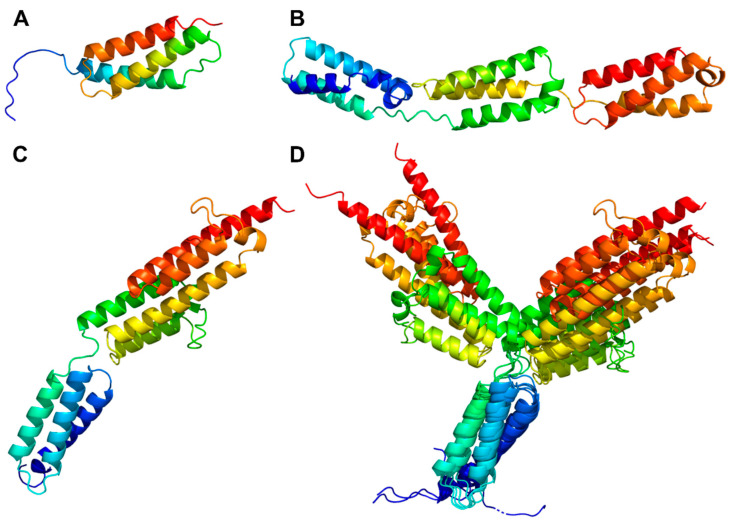
Helix domains (H-domains) of PrtP homologs. The H-domain structures were derived from AlphaFold 2 models of PrtH (**A**), PrtS (**B**), PrtP_MS22337_ (**C**), and PrtP_MS22337_, PrtP_MS22333_, PrtP_SK11_, PrtP_Wg2_, and PrtP_NFICC96Q_ (**D**). The latter structures were superimposed on the N-terminal helix region of PrtP_MS22337_ with RMSD values of 0.50–0.76 Å. The non-displayed superimposed structures of the C-terminal helix region had RMSD values of 0.57–0.61 Å. All H-domains were colored from the N-termini (blue) to the C-termini (red).

**Figure 8 microorganisms-11-02256-f008:**
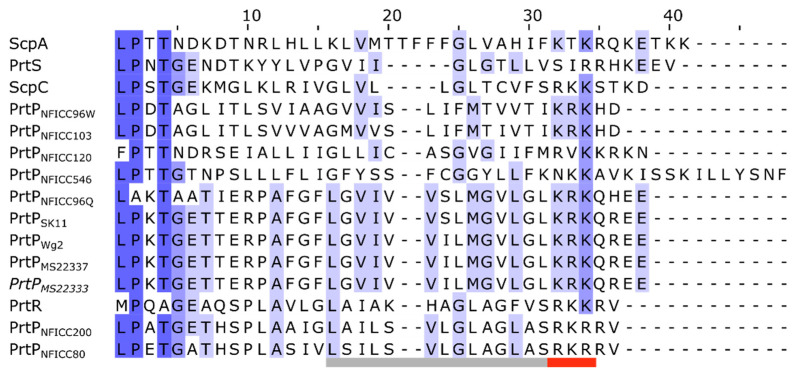
Covalent cell wall attachment domain. The protein sequences of the LPXTG cell wall anchor (AN) domains were aligned for 15 PrtP homologs. Amino acid residues were colored blue with increased sequence identities. The start of the AN-domains was defined with the canonical LPXTG motif or a variant of this motif pattern. Each of the AN-domains also contained a hydrophobic region and a positively charged region, which were highlighted with the gray bar and red bar below the protein sequences, respectively.

**Figure 9 microorganisms-11-02256-f009:**
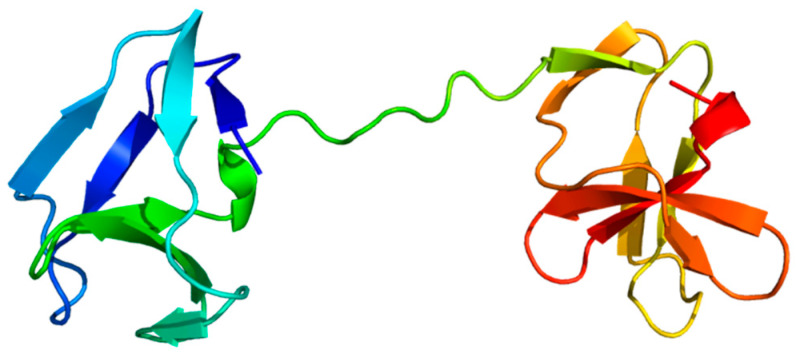
Non-covalent cell wall attachment domain. The surface layer protein A (SlpA) domain of PrtB was modelled by AlphaFold 2 (AF). The domain was colored from the N-terminus (blue) to the C-terminus (red), showing its bipartite structure.

**Table 1 microorganisms-11-02256-t001:** Selected PrtP homologs from databases.

Protease Name	Sequence Accession No.	Structure Model	Species	Bacterial Strain
PrtB	Q48545 ^†^	ma-uvrke ^§^	*Lactobacillus delbrueckii* subsp. *bulgaricus*	NCDO1489
PrtH	Q9S4K2 ^†^	ma-8ai4c ^§^	*Lactobacillus helveticus*	CNRZ32
PrtH2	A4UAD8 ^†^	ma-pes1m ^§^	*Lactobacillus helveticus*	CNRZ32
PrtH3_CNRZ32_	G8DA68 ^†^	-	*Lactobacillus helveticus*	CNRZ32
PrtH3_DSM20075_	EEW67121.1 ^‡^	-	*Lactobacillus helveticus*	DSM 20075
PrtH4	G8DA69 ^†^	-	*Lactobacillus helveticus*	CNRZ32
PrtL	EPB98635.1 ^‡^	ma-sthfc ^§^	*Lactobacillus delbrueckii* subsp. *lactis*	CRL581
PrtS_4F44_	D3KCP7 ^†^	ma-8wf3g ^§^	*Streptococcus thermophilus*	4F44
PrtS_LMD-9_	WP_011681052.1 ^‡^	-	*Streptococcus thermophilus*	LMD-9
PrtP_MS22333_	WWDI00000000 ^‡^	ma-iisw7 ^§^	*Lactococcus lactis*	MS22333
PrtP_MS22337_	WWDK00000000 ^‡^	ma-1x50w ^§^	*Lactococcus lactis*	MS22337
PrtP_NCDO151_	Q02470 ^†^	-	*Lacticaseibacillus paracasei* subsp. *paracasei*	NCDO151
PrtP_SK11_	DQ149245.1 ^‡^	ma-bevya ^§^	*Lactococcus lactis* subsp. *cremoris*	SK11
PrtP_Wg2_	P16271 ^†^	ma-r3cva ^§^	*Lactococcus lactis* subsp. *cremoris*	Wg2
PrtR	Q8GC13 ^†^	ma-i7gkn ^§^	*Lacticaseibacillus rhamnosus*	BGT10
ScpA	P15926 ^†^	3EIF ^¶^	*Streptococcus pyogenes*	B220
ScpC	Q3HV58 ^†^	5XYR ^¶^	*Streptococcus pyogenes*	-
Subtilisin	P00780 ^†^	AF-P00780 ^††^	*Bacillus licheniformis*	-

^†^ UniProtKB. ^‡^ GenBank at NCBI. ^§^ Identifier for AlphaFold 2 (AF) structure models from this paper. The models are available in ModelArchive (modelarchive.org). ^¶^ Identifier for applied protein structure from Protein data bank (PDB). ^††^ Identifiers for applied protein structures from the AlphaFold (AF) protein structure database.

**Table 2 microorganisms-11-02256-t002:** Algorithms used for homology assessment and structure prediction.

Algorithm	Application	Server Address	Reference
AlphaFold 2 and ColabFold	Three-dimensional protein structure modeling	https://github.com/sokrypton/ColabFold (accessed on 15 March 2022)	[22,28]
HHpred	Protein sequence homology detection	https://toolkit.tuebingen.mpg.de/tools/hhpred (accessed on 5 May 2022)	[27]
MAFFT	Multiple protein sequence alignment	https://toolkit.tuebingen.mpg.de/tools/mafft (accessed on 7 September 2022)	[29]
NetSurfP-3.0	Prediction of secondary protein structures and protein intrinsic disorder regions	https://services.healthtech.dtu.dk/service.php?NetSurfP-3.0 (accessed on 23 November 2022)	[30]
SignalP6	Secretory signal peptide and cleavage site prediction	https://services.healthtech.dtu.dk/service.php?SignalP (accessed 5 May 2022)	[26]
T-REKS	Identification of tandem repeat patterns in protein sequences	https://bioinfo.crbm.cnrs.fr/index.php?route=tools&tool=3 (accessed on 9 August 2022)	[31]
XSTREAM	Identification of tandem repeat patterns in protein sequences	https://amnewmanlab.stanford.edu/xstream/ (accessed on 9 August 2022)	[32]

**Table 3 microorganisms-11-02256-t003:** PrtP homologs from lactic acid bacteria with plant origins.

Protease Name	Sequence Accession No. ^†^	Structure Model ^‡^	Species	Bacterial Strain ^§^
PrtP_NFICC1_	OQ263285	-	*Enterococcus durans*	NFICC1
PrtP_NFICC14_	OQ263286	-	*Enterococcus durans*	NFICC14
PrtP_NFICC16_	OQ263287	-	*Enterococcus durans*	NFICC16
PrtP_NFICC18_	OQ263288	-	*Enterococcus durans*	NFICC18
PrtP_NFICC63_	OQ263289	-	*Enterococcus durans*	NFICC63
PrtP_NFICC64_	OQ263290	-	*Enterococcus durans*	NFICC64
PrtP_NFICC80_	OQ263291	ma-8yfr1	*Leuconostoc lactis*	NFICC80
PrtP_NFICC96H_	OQ263292	ma-b49jj	*Leuconostoc pseudomesenteroides*	NFICC96
PrtP_NFICC96Q_	OQ263293	ma-eoqml	*Leuconostoc pseudomesenteroides*	NFICC96
PrtP_NFICC96W_	OQ263294	ma-5kpc0	*Leuconostoc pseudomesenteroides*	NFICC96
PrtP_NFICC99_	OQ263295	-	*Leuconostoc pseudomesenteroides*	NFICC99
PrtP_NFICC102_	OQ263296	-	*Leuconostoc pseudomesenteroides*	NFICC102
PrtP_NFICC103_	OQ263297	ma-u453p	*Pediococcus pentosaceus*	NFICC103
PrtP_NFICC115_	OQ263298	-	*Leuconostoc mesenteroides*	NFICC115
PrtP_NFICC120_	OQ263299	ma-gayjn	*Enterococcus durans*	NFICC120
PrtP_NFICC180_	OQ263300	-	*Leuconostoc pseudomesenteroides*	NFICC180
PrtP_NFICC181_	OQ263301	-	*Leuconostoc pseudomesenteroides*	NFICC181
PrtP_NFICC200_	OQ263302	ma-ioefi	*Leuconostoc mesenteroides*	NFICC200
PrtP_NFICC202_	OQ263303	-	*Leuconostoc pseudomesenteroides*	NFICC202
PrtP_NFICC227_	OQ263304	-	*Leuconostoc pseudomesenteroides*	NFICC227
PrtP_NFICC239_	OQ263305	-	*Leuconostoc mesenteroides*	NFICC239
PrtP_NFICC240_	OQ263306	-	*Leuconostoc mesenteroides*	NFICC240
PrtP_NFICC313_	OQ263307	-	*Leuconostoc pseudomesenteroides*	NFICC313
PrtP_NFICC322_	OQ263308	-	*Leuconostoc pseudomesenteroides*	NFICC322
PrtP_NFICC323_	OQ263309	-	*Leuconostoc pseudomesenteroides*	NFICC323
PrtP_NFICC476_	OQ263310	-	*Carnobacterium maltaromaticum*	NFICC476
PrtP_NFICC546_	OQ263311	ma-ytnir	*Carnobacterium maltaromaticum*	NFICC546

^†^ GenBank at NCBI accession numbers for the sequences from this study. ^‡^ Identifier for AlphaFold 2 (AF) structure models from this paper. The models are available in ModelArchive (modelarchive.org). ^§^ Bacterial strains from the National Food Institute Culture Collection (NFICC) at the Technical University of Denmark.

## Data Availability

TNew PrtP sequence data of this study is accessible at GenBank at NCBI, and protein structure models are accessible at ModelArchive (modelarchive.org).

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
