# Peer review of "Comparative Structure Analysis of the Multi-Domain, Cell Envelope Proteases of Lactic Acid Bacteria"

_microorganisms, 2023, doi:10.3390/microorganisms11092256_

Round 1
Reviewer 1 Report
The genesis of this manuscript is a comparative analysis of LAB proteinases based on an in silico approach. The authors conclude that phylogenetic clusters of PrtP homologs have diverse domain structures that do not necessarily correlate with the natural source of the bacterial strain which can be explained by HGT. Furthermore, the PrtP homologs contain different combinations of Fn type-III-like domains that are divided, based on structure, into twelve groups, among which Fn1 and Fn2 show the greatest conservation. The authors also found the W-domain, which is likely involved in various protein interactions. A way to test the overall results would be through in vitro evolution. The title and abstract are appropriate for the content of the text. Furthermore, the article is well constructed, the experiments were well conducted, and the analysis was well performed.
I have a few general suggestions for the improvement:
- shorten the introduction part or even the results and discussion section by introducing more straightforward statements to improve the readability of the text
- avoid using nonspecific terminology, such as adhesive platform
- Since recently a novel taxonomy of species of the Lactobacillus group is introduced, I suggest updating accordingly. e.g. Lactobacillus casei was reclassified as Lacticaseibacillus casei
- it would also be attractive to briefly explain the application potential in the discussion
The title and abstract are appropriate for the content of the text. Furthermore, the article is well constructed, the experiments were well conducted, and the analysis was well performed.
Author Response
Comments on the review report (Reviewer 1)
Thank you for the positive feedback and suggestions for improvements.
- Shorten the introduction part or even the results and discussion section by introducing more straightforward statements to improve the readability of the text.
The introduction and discussion sections have been shorten to clarify the statements of the manuscript and improve readability.
- Avoid using nonspecific terminology.
The use of adhesive platform has been removed from the text.
- Update according to recent taxonomy of Lactobacillus
Species names have been corrected to recent taxonomy in the text. The figures and tables have also been checked. Following changes of names have been made:
- Lactobacillus paracasei changed to Lacticaseibacillus paracasei
- Lactobacillus rhamnosus changed to Lacticaseibacillus rhamnosus
- Briefly explain the application potential in the discussion
The application potentials of the main findings have been clarified through the discussion section.
Reviewer 2 Report
General Comments of Reviewer to Author
The subject of the article is very important and would be interesting for many readers. The authors have provided comprehensive information about the potential of their study and its application. The title also gives a general overview of the entire review manuscript.
The authors need to address all sections highlighted with comments provided in the draft manuscript. The abstract is concise and provides great insight on the subject matter as outlined in the draft manuscript.
The materials and methods section has been well elaborated with an excellent experimental design
The results, discussion and conclusion sections have been well explained for clarity and understanding.
There are also some abbreviations in the text that have not been appropriately defined. It is best practice to first define abbreviations the first time that they are used.

Author Response
Comments on the review report (Reviewer 2)
Thank you for the positive feedback and suggestions for improvements.
We have made changes according to the highlighted sections with comments in the draft manuscript, addressing following issues:
- Cite references. The citations of references have been improved by adding references in highlighted sections.
- Correcting species name of Lactobacillus.
The misspelling Lactobacillus delbreecki has been corrected to Lactobacillus delbrueckii in the text.
- Family and species names of the bacteria and et al. have been corrected and written in
- Double spaces are removed.
- Abbreviations in the text have been checked. However, CLC is a name in CLC Main Workbench.